# *FHL2* Inhibits SARS-CoV-2 Replication by Enhancing *IFN-β* Expression through Regulating *IRF-3*

**DOI:** 10.3390/ijms25010353

**Published:** 2023-12-26

**Authors:** Zhiqiang Xu, Mingyao Tian, Qihan Tan, Pengfei Hao, Zihan Gao, Chang Li, Ningyi Jin

**Affiliations:** 1Agricultural College, Yanbian University, Yanji 133002, China; xzq1660@163.com (Z.X.);; 2Research Unit of Key Technologies for Prevention and Control of Virus Zoonoses, Chinese Academy of Medical Sciences, Changchun Veterinary Research Institute, Chinese Academy of Agricultural Sciences, Changchun 130122, China; klwklw@126.com (M.T.);

**Keywords:** SARS-CoV-2, *FHL2*, *IFN*-*β*, *IRF-3*, infection

## Abstract

SARS-CoV-2 triggered the global COVID-19 pandemic, posing a severe threat to public health worldwide. The innate immune response in cells infected by SARS-CoV-2 is primarily orchestrated by type I interferon (IFN), with *IFN-β* exhibiting a notable inhibitory impact on SARS-CoV-2 replication. *FHL2*, acting as a docking site, facilitates the assembly of multiprotein complexes and regulates the transcription of diverse genes. However, the association between SARS-CoV-2 and *FHL2* remains unclear. In this study, we report for the first time that SARS-CoV-2 infection in Caco2 cells results in the upregulation of *FHL2* expression, while the virus’s N proteins can enhance *FHL2* expression. Notably, the knockdown of *FHL2* significantly amplifies SARS-CoV-2 replication in vitro. Conversely, the overexpression of *FHL2* leads to a marked reduction in SARS-CoV-2 replication, with the antiviral property of *FHL2* being independent of the cell or virus type. Subsequent experiments reveal that *FHL2* supports *IFN-β* transcription by upregulating the expression and phosphorylation of IRF-3, thereby impeding SARS-CoV-2 replication in cells. These findings highlight *FHL2* as a potential antiviral target for treating SARS-CoV-2 infections.

## 1. Introduction

Severe acute respiratory syndrome coronavirus 2 (SARS-CoV-2) is a positive-sense single-stranded RNA virus with an envelope which is classified under β coronavirus group 2 [1]. Its genome encompasses 14 open reading frames (ORFs), two-thirds of which encode 16 non-structural proteins (NSP 1–16) constituting the replicase complex [2,3]. The remaining ORFs encode nine accessory proteins and four structural proteins: spike (S), envelope (E), membrane (M), and nucleocapsid (N), with spike facilitating SARS-CoV-2 entry into host cells [4]. Following the emergence of SARS-CoV and MERS-CoV, SARS-CoV-2 represents the third zoonotic human coronavirus in this century [5]. SARS-CoV-2 efficiently infects the upper respiratory tract (URT) [6] and the intestine [7], resulting in pneumonia characterized by fever, dry cough, and occasionally gastrointestinal symptoms.

The spike protein of SARS-CoV-2 features a receptor-binding domain (RBD) that directly interacts with the cellular receptor angiotensin-converting enzyme 2 (ACE2) and an S1/S2 multivalent cleavage site cleaved by cellular cathepsin L and transmembrane protease serine 2 (TMPRSS2) [8,9,10]. SARS-CoV-2 binds to and enters cells through ACE2 and TMPRSS2 on the cell surface. Upon RNA release from endosomes, it is recognized by retinoic acid-inducible gene I (RIG-I)-like receptors (RLRs), including RIG-I, melanoma differentiation gene 5 (MDA5), and toll-like receptors (TLRs) [11]. RIG-I and MDA5 activate the downstream adaptor mitochondrial antiviral signaling protein (MAVs) on mitochondria. MAVs subsequently recruit two kinases, IKKε and tank-binding kinase 1 (TBK1), leading to the phosphorylation and nuclear translocation of IFN regulatory factor 3 (IRF3), inducing the expression and secretion of interferon (IFN) [12]. As the primary defense against viruses, type I interferons (IFNs) play a crucial role in initiating the host antiviral response. Previous studies indicate that *IFN-β* is the most effective subtype against human coronavirus infection [13,14,15]. SARS-CoV-2 employs various strategies to interfere with key host signaling factors, antagonizing the IFN system through specific proteins, including accessory proteins, non-structural proteins, and structural proteins [16]. The literature reveals that over half of SARS-CoV-2 proteins have antagonistic effects on the interferon response by targeting viral sensors or obstructing downstream antiviral signaling molecules [17,18,19,20,21,22,23].

Four and a half LIM domains 2 (*FHL2*) belongs to the LIM-only protein family. The LIM domain, containing two cysteine-rich zinc finger-like interaction sequences, is regarded as a docking site facilitating the assembly of multiprotein complexes [24]. Functioning as a scaffold protein, *FHL2* can regulate the structure, activity, and function of its interaction partners. By binding to target proteins, *FHL2* modulates signal transduction pathways and subsequent gene regulation [25,26,27,28,29,30]. The expression and function of *FHL2* have been extensively studied in various diseases, including different types of cancer [31,32,33], cardiovascular disease [25,34,35], and overall metabolism [36]. Recent research has unveiled *FHL2*’s crucial role in innate immune-related pathways. In the toll-like receptor (TLR)/tumor necrosis factor (TNF) signaling pathway, *FHL2* interacts with tumor necrosis factor receptor-related factor 2 (TRAF2), TRAF4, and TRAF6, thereby activating the NF-κB signaling pathway [37]. *FHL2* can also regulate interleukin-6 expression in muscle cells through the p38 MAPK-mediated NF-κB pathway [38]. Moreover, in the nucleus, it directly binds to members of the p300/cbp family of transcriptional coactivators [39]. Previous studies have illustrated that *FHL2* is a crucial regulator of the innate cellular immune response to influenza A virus (IAV) infection, enhancing IRF-3-dependent transcription of the *IFN-β* gene to inhibit virus replication in cells [40].

While some studies have demonstrated that *FHL2* can enhance virus-dependent induction of *IFN-β*, the specific mechanism requires further elucidation. In this study, we observed an association between the protein level of *FHL2* and the infection of Caco2 cells with SARS-CoV-2. Knockdown and overexpression experiments conducted in Caco2 and 293T-ACE2 cells revealed that *FHL2* inhibited the replication of SARS-CoV-2. Further experiments indicated that *FHL2* plays a pivotal role in innate immunity by promoting the expression and phosphorylation of IRF-3, thereby enhancing the transcription of *IFN-β*.

## 2. Results

### 2.1. Infection of Caco2 Cells with SARS-CoV-2 Resulted in Increased FHL2 Protein Levels

By analyzing proteomic data (IPX0003647000) [41], a total of 199 differential proteins were identified at 12hpi compared to 0hpi (Figure 1A). KEGG pathway enrichment analysis was conducted on the differential proteins (Figure 1B), revealing particular interest in the P53 signaling pathway and the peroxisome proliferator-activated receptor (PPAR) signaling pathway, both associated with inflammation and immunity [42,43]. Consequently, we speculated that among these proteins, there might be those with antiviral potential. To broaden our screening, we referred to proteins in P53-related pathways (transcription P53 signaling pathway, P53 signaling, P53 pathway) and PPAR-related pathways (PPARA activates gene expression, PPAR signaling pathway) in the PathCards database (Appendix A). Upon screening the proteins in the aforementioned pathways, among the differential proteins, *FHL2* and RELA (RELA proto-oncogene) were found to meet the screening conditions simultaneously (Figure 1C, Appendix A). The fold change in *FHL2* was higher than that of RELA (Appendix A). Consequently, our focus was directed towards *FHL2*.

Caco2 cells were infected with SARS-CoV-2 at 0.01 MOI, and the infected cells were collected at 12 hpi, 24 hpi, and 36 hpi. The protein level of *FHL2* was assessed through WB. The results demonstrated that the change trend in the *FHL2* protein level was consistent with the proteomic data (Figure 1E,F). qRT-PCR results also confirmed the change trend in *FHL2* in terms of the mRNA expression level (Figure 1G). Subsequently, we delved into how SARS-CoV-2 promoted the expression of *FHL2*. It has been reported that the SARS-CoV-2 N protein can trigger NF-κB-mediated inflammatory responses in cells [44]. *FHL2* can also regulate the inflammatory response through NF-κB [38]. Therefore, we evaluated the effect of the N protein on *FHL2* expression by expressing it in Caco2 cells. WB results indicated that intracellular expression of the N protein led to an increase in the *FHL2* expression level (Figure 1H). This finding suggested that the viral N protein could elevate the expression of *FHL2* in Caco2 cells infected by SARS-CoV-2.

### 2.2. Knockdown of FHL2 Promotes SARS-CoV-2 Proliferation

To assess the impact of *FHL2* on SARS-CoV-2 infection, shRNA was employed to downregulate *FHL2* expression in Caco2 cells and 293T-ACE2 cells. Western blot (WB) results demonstrated a marked reduction in *FHL2* expression upon knockdown (Figure 2A,B). Subsequently, cells were infected with different strains (WT strain, BA.1 strain) at 0.01 MOI and 0.1 MOI, respectively. The protein levels of SARS-CoV-2 N protein were assessed through WB at 12 h and 24 h post-infection. The findings revealed that *FHL2* knockdown heightened the N protein expression, indicating that diminished *FHL2* levels promoted the replication of both strains in distinct cell lines (Figure 2C,D).

Simultaneously, the viral replication level in the supernatant was quantified using qRT-PCR. The results indicated that *FHL2* knockdown led to an augmentation in the RNA copies of the N protein for both strains in the supernatant (Figure 2E,F). Subsequent determination of virus titers through TCID50 also showed a significant increase in the progeny virus titer upon *FHL2* knockdown (Figure 2G). In conclusion, the knockdown of *FHL2* was observed to enhance the replication of two SARS-CoV-2 strains (WT strain, BA.1 strain) in both Caco2 cells and 293T-ACE2 cells.

### 2.3. Overexpression of FHL2 Inhibits SARS-CoV-2 Proliferation

To comprehensively assess the impact of *FHL2* on SARS-CoV-2 proliferation, we introduced *FHL2* overexpression in both Caco2 cells and 293T-ACE2 cells. Western blot (WB) analysis confirmed the abundant expression of *FHL2* in the cells (Figure 3A,B). Subsequently, the cells were infected with various strains at 0.01 MOI and 0.1 MOI for 12 h and 24 h, respectively, and the N protein expression level was determined through WB. The findings revealed that the overexpression of *FHL2* led to a reduction in N protein expression, signifying that *FHL2* overexpression restrained virus proliferation in the cells (Figure 3C,D) and that this inhibitory effect was dose-dependent (Figure 3E). The virus replication in the supernatant was quantified using qRT-PCR, demonstrating that *FHL2* overexpression resulted in diminished viral load in the supernatant (Figure 3F,G). These outcomes affirmatively corroborate that *FHL2* possesses the capability to impede SARS-CoV-2 replication.

### 2.4. FHL2 Promotes IFN-β Transcription

*IFN-β* plays a pivotal role in suppressing SARS-CoV-2 replication [45]. Simultaneously, research indicates that *FHL2* can enhance *IFN-β* transcription in the A549 cell line [40]. Consequently, we sought to confirm whether *FHL2* exerts a similar regulatory influence on *IFN-β* in Caco2 cells. Following *FHL2* knockdown, cells were infected with 1 MOI of the WT strain, and qRT-PCR analysis conducted after 6 h revealed a substantial reduction in *IFN-β* mRNA levels upon *FHL2* depletion (Figure 4A). Conversely, upon *FHL2* overexpression and subsequent SARS-CoV-2 inoculation, qRT-PCR results demonstrated an augmentation in *IFN-β* transcription (Figure 4B).

The accumulation of viral RNA in infected cells triggers the expression and secretion of *IFN-β* [46]. Therefore, employing RNA extracted from the supernatant of SARS-CoV-2-infected cells as a stimulus and the supernatant of uninfected cells as a control, qRT-PCR results illustrated that *FHL2* knockdown hindered *IFN-β* transcription (Figure 4C). In contrast, *FHL2* overexpression facilitated IFN transcription (Figure 4D). These findings align with the outcomes observed when SARS-CoV-2 stimulated Caco2 cells. Hence, *FHL2* promotes *IFN-β* transcription in Caco2 cells, signifying that *FHL2* can impede SARS-CoV-2 replication by enhancing *IFN-β* transcription.

### 2.5. FHL2 Supports the Expression and Phosphorylation of IRF-3

IRF-3 is a pivotal transcription factor for *IFN-β* [47]. Hence, we investigated the impact of *FHL2* on IRF-3. Knockdown and overexpression of *FHL2* were performed in Caco2 cells, and the expression level of IRF-3 was assessed through WB and qRT-PCR. The outcomes revealed that *FHL2* knockdown significantly diminished the protein and mRNA levels of IRF-3 in Caco2 cells (Figure 5A,B), while *FHL2* overexpression augmented both the protein and mRNA levels of IRF-3 in Caco2 cells (Figure 5C,D). Consequently, *FHL2* can enhance IRF-3 expression in Caco2 cells. The activation of IRF-3 involves phosphorylation, leading to its nuclear translocation, where it binds to the *IFN-β* promoter region to facilitate transcription. To explore this further, we examined the effect of *FHL2* on IRF-3 phosphorylation and nuclear translocation. Following *FHL2* knockdown, Caco2 cells were infected with the SARS-CoV-2 WT strain at 1 MOI for 6 h, and the phosphorylation of IRF-3 was assessed through WB. The results indicated that *FHL2* knockdown resulted in reduced IRF-3 phosphorylation (Figure 5E). Furthermore, IRF-3 phosphorylation increased upon *FHL2* overexpression (Figure 5F). Caco2 cells overexpressing *FHL2* were stimulated with viral RNA, and the levels of IRF-3 in the nucleus, cytoplasm, and whole cell were examined through WB. The findings demonstrated that *FHL2* overexpression significantly elevated IRF-3 levels in the nucleus (Figure 5G). In conclusion, *FHL2* has the capacity to upregulate the expression and phosphorylation of IRF-3, thereby enhancing the transcription of *IFN-β* to exert antiviral function (Figure 6).

## 3. Discussion

SARS-CoV-2 has been prevalent worldwide since 2019 [48]. Numerous research teams have conducted extensive investigations on the virus in recent years. However, there is still considerable room for improvement in treatment [49]. Therefore, it is imperative to study the effects of SARS-CoV-2 on human cell signaling pathways and the impact of cellular intrinsic proteins on SARS-CoV-2 replication.

*FHL2* is a pivotal transcription factor in cells, and its mechanisms have been extensively studied [31,32,33,34,35,36,37,38,39]. Recent research has highlighted its crucial role in immune and inflammation-related pathways. This study specifically focuses on the impact of *FHL2* on the innate immune pathway, providing a partial analysis of its antiviral mechanism.

In this investigation, we explored the relationship between SARS-CoV-2 and *FHL2*, discovering that the N proteins of SARS-CoV-2 could activate the expression of *FHL2* in cells. Previous studies have indicated that *FHL2* is a significant regulator of the anti-influenza A virus response by enhancing the innate immune response of infected cells [40]. Our study validated the effect of *FHL2* on the replication of SARS-CoV-2 in cells, reinforcing the broad-spectrum antiviral nature of *FHL2*.

Subsequently, we delved into the mechanism by which *FHL2* inhibits SARS-CoV-2 replication. As a crucial component of innate immunity, type I interferons play a pivotal role in the early response to viral infections, especially respiratory viruses [50]. SARS-CoV-2 infection can lead to varying degrees of clinical symptoms, with both mild and severe infections accompanied by a type I interferon response [51]. Our experimental findings demonstrated that *FHL2* could enhance the transcription of *IFN-β*. Early use of IFN following a novel coronavirus infection has a protective effect. At later time points, when IFN and inflammatory cytokines become pathogenic, inhibiting IFN and cytokine signaling could be an effective therapeutic option to restore the balance of an excessive immune response [51]. The expression of *IFN-β* can be regulated by manipulating *FHL2* expression. Hence, *FHL2* can serve as a target, offering more therapeutic options for treating SARS-CoV-2.

Previous studies have indicated that *FHL2* promotes the transcription of IRF-3, aligning with the results of this study [40]. Here, we discovered that *FHL2* promotes the expression of IRF-3, influencing its nuclear translocation and phosphorylation. IRF-3 is indispensable for *IFN-β* production upon viral infection, further emphasizing the significant role of *FHL2* in regulating *IFN-β* responses. While previous studies suggested that *FHL2* does not affect the phosphorylation of IRF-3 but rather enhances its function after entering the nucleus [40], our findings differ, potentially influenced by distinct cell lines and virus species.

## 4. Materials and Methods

### 4.1. Cell Culture and Transfection

Human colon carcinoma Caco2 cells, 293T-hACE2 cells, and Vero-E6 cells were cultured in Dulbecco’s Modified Eagle Medium (DMEM) supplemented with 10% fetal bovine serum (FBS) and 1% penicillin/streptomycin at 37 °C with 5% CO_2_. Cells were seeded in 24-well, 12-well, and six-well plates and transfected using TransIT-X2^®^ transfection reagent (Mirus Bio LLC, Madison, WI, USA) following the manufacturer’s instructions.

### 4.2. Viruses

Authentic SARS-CoV-2 WT (IME-BJ01 strain, GenBank No. MT291831) and Omicron BA.1 (SARS-CoV-2 strain Omicron CoV/human/CHN_CVRI-01/2022) strains were isolated from COVID-19 patients. All virus experiments were conducted in a Biosafety Level 3 laboratory with standard operating procedures.

### 4.3. Virus Infection

Cells at approximately 90% confluence were washed with phosphate-buffered saline and inoculated with SARS-CoV-2 at a multiplicity of infection (MOI). After 1 h of adsorption, the inocula were removed, and cells were maintained in medium containing 2% FBS at 37 °C in a 5% CO_2_ incubator for an indicated time. Mock infected cells were generated using culture medium as the control inoculum.

### 4.4. Plasmid Construction

A reverse transcription polymerase chain reaction (RT-PCR) was used to generate cDNA for *FHL2* (GenBank no. 2274), M (GenBank no. 43740571), and N (GenBank no. 43740575). *FHL2* was subcloned into the eukaryotic expression vector pcDNA3.1 with a His-tag at its C-terminal. M and N were subcloned into the eukaryotic expression vector pcDNA3.1 with the Strep tag at their C-terminal. Target genes were verified through DNA sequencing.

### 4.5. shRNA

Short hairpin RNA (shRNA) specifically inhibiting *FHL2* was synthesized by Ribobio Co (Guangzhou, China) based on the *FHL2* gene sequence (GenBank no. 2274).

### 4.6. Nuclear and Cytoplasmic Extraction

In a six-well plate, 60–70% confluent Caco2 cells were transfected with the indicated plasmids for 24 h, then stimulated by viral RNA. Nuclear and cytoplasmic extraction were performed using reagents according to the manufacturer’s instructions (Thermo Fisher Scientific, Waltham, MA, USA). The process involved adding cytoplasmic lysate CER I to resuspend the cells, adding cytoplasmic lysate CER II, and subsequent centrifugation to obtain the cytoplasmic extract. The nuclear extract was obtained by adding nuclear lysate NER to the insoluble pellet, followed by centrifugation. All lysis processes were performed on ice.

### 4.7. Western Blot (WB)

Cells were harvested, lysed, and their protein concentrations determined using a bicinchoninic acid (BCA) protein assay kit (Beyotime, Shanghai, China). Subsequently, proteins were combined with a loading buffer and denatured through boiling. Total cellular extracts were prepared and separated via 10% SDS-PAGE. The proteins were then transferred onto a PVDF membrane (GE Healthcare, Chicago, IL, USA). Following SDS-PAGE, the transfer to the PVDF membrane, and 2 h of room temperature blocking, the corresponding primary antibody was added and incubated overnight at 4 °C. Subsequently, the membrane underwent a 50 min incubation with either goat anti-rabbit or goat anti-mouse secondary antibody.

The primary antibodies employed in this study included the *FHL2* polyclonal antibody, GAPDH polyclonal antibody, alpha tubulin monoclonal antibody, IRF-3 polyclonal antibody, phospho-IRF-3 (Ser396) polyclonal antibody, His-tag monoclonal antibody, LaminA/C Polyclonal Antibody (Wuhan Sanying, Wuhan, China), Strep Tag II Monoclonal Antibody (Thermo Fisher Scientific, Waltham, MA, USA), and SARS-CoV-2 Nucleocapsid Polyclonal Antibody (Cell Signaling Technology, Danvers, MA, USA).

### 4.8. Quantitative Real-Time PCR

Viral RNA in the supernatant was extracted using the QIAamp^®^ Viral RNA Mini kit (Qiagen, Hilden, NRW, Germany) following the manufacturer’s instructions. qPCR was performed employing the HiScript II U+ One Step qRT-PCR Probe Kit (Vazyme, Nanjing, China). Primers and probes were designed based on the viral genomic RNA gene sequences of SARS-CoV-2 and were procured from Sangon Biotech Co.; Ltd. (Shanghai, China). The viral RNA load was determined using TaqMan quantitative real-time PCR, as previously outlined [52].

Intracellular RNA was purified according to the manufacturer’s instructions (Life Technologies, Carlsbad, CA, USA). Total RNA (500 ng) served as a template and underwent reverse transcription to cDNA using M–MLV reverse transcriptase (Promega, Madison, WI, USA). RT-qPCR was conducted using SYBR Green Master Mix (TOYOBO, Osaka, Japan) on a BIO–RAD CFX96 Real-Time PCR System (BIO–RAD, Singapore). For relative quantitation analysis, samples were normalized to the expression of the housekeeping gene glyceraldehyde–3–phosphate dehydrogenase (GAPDH). The specific primers used are detailed in Table 1.

### 4.9. TCID50

Vero-E6 cells were plated into 96-well plates at a density of 1 × 10^5^ cells per well, and the sample was continuously diluted by ten-fold (100 µL/well), with three repetitions for each sample. After the cells exhibited obvious cytopathic effects (CPE), the number of CPE holes under each dilution was recorded, and TCID50 was calculated using the Reed–Muench method.

### 4.10. Bioinformatic Analyses

Proteomic data (IPX0003647000) were acquired from iProX for subsequent bioinformatic investigations. Mass spectrometry data extraction was conducted using maxquant (v1.6.15.0). The retrieval parameters included the Homo sapiens 9606 SP 20201214.fasta database (20395 sequences), with the addition of a reverse database to compute the false positive rate (FDR) attributable to random matches. P values were calculated using Student’s *t*-test, and the Benjamini–Hochberg method was applied for P value correction. Instances where Padj ≤ 0.05 were considered significant, with change thresholds of more than 1.5 for significant upregulation and less than 1/1.5 for significant downregulation serving as the criteria for screening differential proteins. Enrichment analysis of Kyoto Encyclopedia of Genes and Genomes pathways was carried out using the clusterProfiler package in R (4.0.4) with default parameters [53].

### 4.11. Statistical Analyses

Unless explicitly stated otherwise, all experiments underwent replication a minimum of three times. Data were presented as means ± standard deviation (SD). Statistical analyses were performed using GraphPad Prism 9 software. The presented micrographs represent typical images from three distinct experiments, as denoted by the figure legends. Unpaired *t*-tests were employed to determine *p*-values between two groups for quantitative data in the specified experiments. Significance was attributed to *p*-values less than 0.05 in the comparison between the two groups.

## 5. Conclusions

In summary, cellular infection by SARS-CoV-2 induces an elevation in the expression level of *FHL2*. This heightened expression of *FHL2* demonstrates the potential to impede the replication of SARS-CoV-2, either through inhibitory mechanisms or by fostering the transcription of *IFN-β*. Notably, *FHL2* is implicated in the promotion of *IFN-β* transcription, a process achieved by augmenting the expression and phosphorylation of IRF-3.

## Figures and Tables

**Figure 1 ijms-25-00353-f001:**
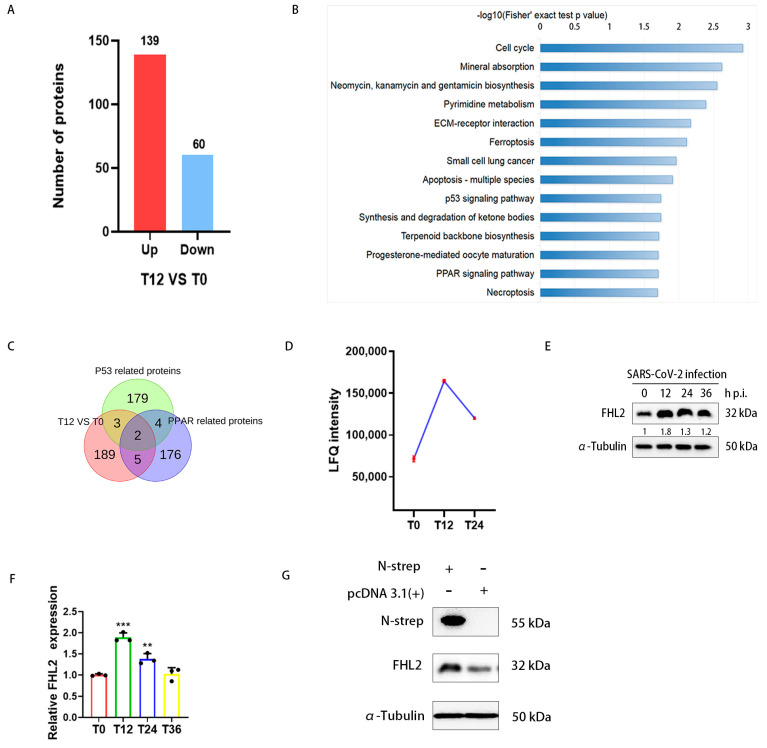
The proteomic analysis revealed that SARS-CoV-2 modulates *FHL2* expression in cells. Caco2 cells were infected with the WT strain at 0.01 MOI, and the infected cells were collected at 0, 12, and 24 h post-infection (hpi), followed by lysate preparation for quantitative proteomic analysis. (**A**) The histogram illustrates differential proteins at 12 hpi compared to 0 hpi. Under the condition of Padj ≤ 0.05, 2977 proteins were screened, with 139 proteins up-regulated and 60 proteins down-regulated (Appendix A). (**B**) KEGG analysis of differentially expressed proteins, highlighting the significant enrichment of the first 14 pathways in differentially expressed proteins (*p* < 0.05). (**C**) Venn diagram depicting proteins related to the P53 signaling pathway and PPAR signaling pathway selected from the differential proteins (Appendix A). (**D**) Graph showing the change trend in *FHL2* LFQ intensity in Caco2 cells infected with SARS-CoV-2. (**E**) Western blot analysis of *FHL2* protein levels in Caco2 cells infected with SARS-CoV-2 at 0.01 MOI at 12 hpi, 24 hpi, and 36 hpi. (**F**) RT-qPCR results illustrating the mRNA levels of *FHL2* in Caco2 cells infected with SARS-CoV-2 at 0.01 MOI at 12 hpi, 24 hpi, and 36 hpi. (**G**) Western blot analysis of *FHL2* expression levels in Caco2 cells transfected with SARS-CoV-2 N protein expression plasmids for 24 h. Three biological replicates were performed for each sample. **, *p* < 0.01; ***, *p* < 0.001.

**Figure 2 ijms-25-00353-f002:**
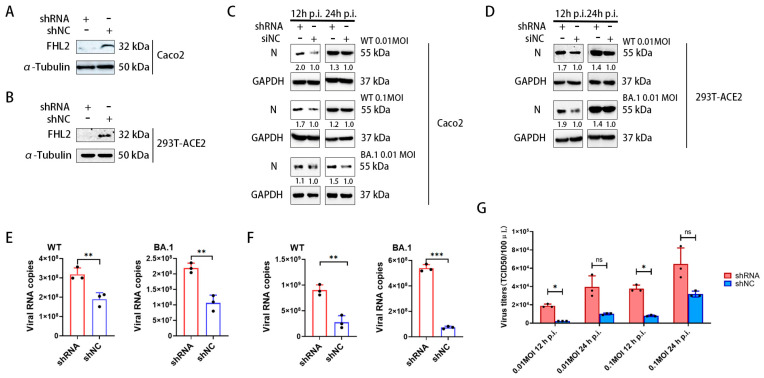
The knockdown of FHL2 facilitates SARS-CoV-2 proliferation. (**A**,**B**) WB analysis of FHL2 protein levels following the transfection of shRNA into Caco2 cells and 293T-ACE2 cells for 48 h. (**C**) Upon FHL2 knockdown in Caco2 cells, the cells were infected with WT strain at 0.01 MOI and 0.1 MOI and BA.1 strain at 0.01 MOI, respectively. Viral N protein levels were assessed through WB at 12 h and 24 h post-infection. (**D**) Following FHL2 knockdown in 293T-ACE2 cells, the cells were infected with WT strain at 0.01 MOI and 0.1 MOI and BA.1 strain at 0.01 MOI, respectively. Viral N protein levels were examined through WB at 12 h and 24 h post-infection. (**E**) After FHL2 knockdown in Caco2 cells infected with WT strain at 0.01 MOI and BA.1 strain at 0.01 MOI, the RNA copies of the N protein were measured through qRT-PCR at 12 h post-infection. (**F**) Following FHL2 knockdown in 293T-ACE2 cells and infection with WT strain at 0.01 MOI and BA.1 strain at 0.01 MOI, the RNA copies of the N protein were quantified through qRT-PCR at 12 h post-infection. (**G**) After FHL2 knockdown in Caco2 cells and infection with WT strain at 0.01 MOI and 0.1 MOI, the TCID50 was determined at 12 h and 24 h post-infection. Three biological replicates were conducted for each sample. *, *p* < 0.05; **, *p* < 0.01; ***, *p* < 0.001, ns, no significant difference.

**Figure 3 ijms-25-00353-f003:**
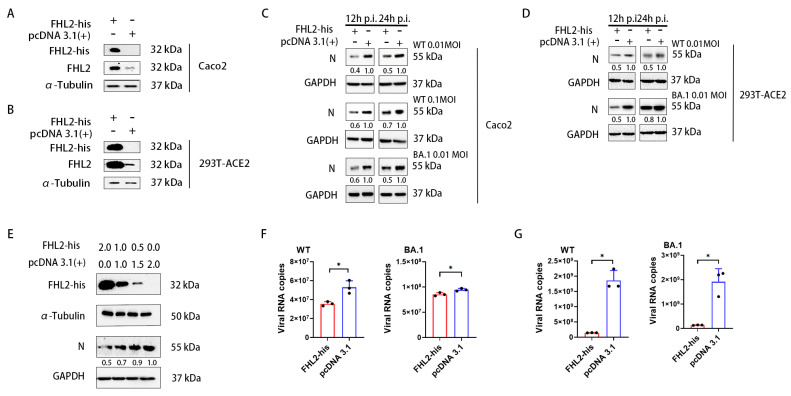
Elevated *FHL2* expression hinders SARS-CoV-2 proliferation. (**A**,**B**) Western blot analysis of *FHL2* protein levels following transfection with the *FHL2*-his expression plasmid in Caco2 cells and 293T-ACE2 cells for 24 h. (**C**) Upon overexpression of *FHL2* in Caco2 cells, the cells were infected with 0.01 MOI, 0.1 MOI WT strain, and 0.01 MOI BA.1 strain, respectively, and the viral N protein was assessed through WB at 12 hpi and 24 hpi. (**D**) Following *FHL2* overexpression in 293T-ACE2 cells, the cells were infected with 0.01 MOI, 0.1 MOI WT strain, and 0.01 MOI BA.1 strain, respectively, and the viral N protein was monitored through WB at 12 hpi and 24 hpi. (**E**) The *FHL2*-his expression plasmid was transfected into cells with 2 μg, 1 μg, and 0.5 μg; then, 24 h later, the cells were infected with the WT strain at 0.01 MOI and the viral N protein was evaluated through WB at 12 hpi. (**F**) Upon *FHL2* overexpression in Caco2 cells, the cells were infected with 0.01 MOI WT strain and 0.01 MOI BA.1 strain, respectively, and the RNA copies of the N protein were determined through qRT-PCR at 12 hpi. (**G**) After *FHL2* overexpression in 293T-ACE2 cells, the cells were infected with 0.01 MOI WT strain and 0.01 MOI BA.1 strain, respectively, and the RNA copies of the N protein were quantified through qRT-PCR at 12 hpi. Three biological replicates were conducted for each sample. *, *p* < 0.05.

**Figure 4 ijms-25-00353-f004:**
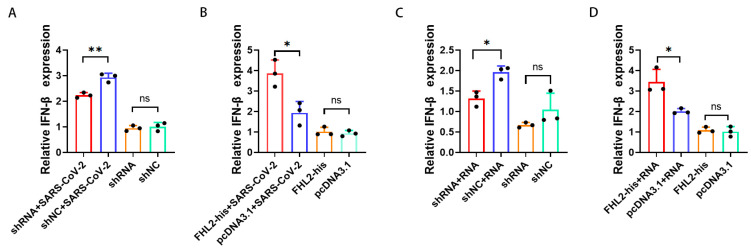
*FHL2* Promotes *IFN-β* Transcription. (**A**) *FHL2* was depleted in Caco2 cells, followed by infection with 1 MOI of SARS-CoV-2 for 6 h, and qRT-PCR was employed to assess *IFN-β* mRNA levels. (**B**) *FHL2* was overexpressed in Caco2 cells, subsequently infected with 1 MOI of SARS-CoV-2 for 6 h, and *IFN-β* mRNA levels were evaluated using qRT-PCR. (**C**) *FHL2* was knocked down in Caco2 cells, and the cells were stimulated with SARS-CoV-2 RNA for 12 h, with qRT-PCR used to measure *IFN-β* mRNA levels. (**D**) *FHL2* was overexpressed in Caco2 cells, and the cells were stimulated with SARS-CoV-2 RNA for 12 h, followed by qRT-PCR to determine *IFN-β* mRNA levels. Three biological replicates were conducted for each sample. *, *p* < 0.05; **, *p* < 0.01, ns, no significant difference.

**Figure 5 ijms-25-00353-f005:**
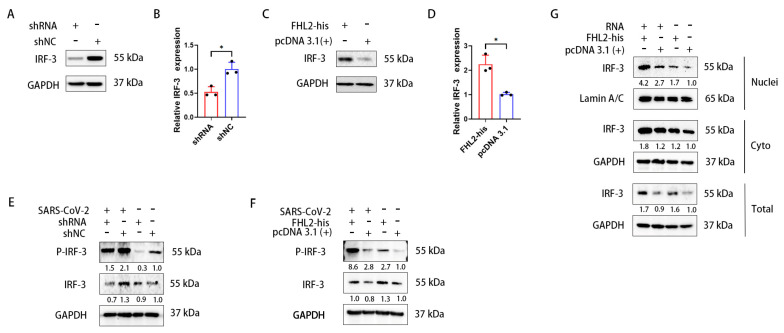
*FHL2* promotes the expression of IRF-3 and its phosphorylation. (**A**,**B**) WB and qRT-PCR analyses revealed the protein and mRNA levels of IRF-3 after 48 h of shRNA transfection in Caco2 cells. (**C**,**D**) WB and qRT-PCR analyses showed the protein and mRNA levels of IRF-3 after 24 h of transfection of the *FHL2*-his expression plasmid in Caco2 cells. (**E**,**F**) *FHL2* was knocked down and overexpressed in Caco2 cells, respectively, and the cells were infected with 1 MOI of SARS-CoV-2 for 6 h before WB detection of IRF-3 and phosphorylated IRF-3. (**G**) After overexpression of *FHL2* in Caco2 cells, the cells were stimulated with SARS-CoV-2 RNA for 12 h. The protein levels of IRF-3 in the nucleus, cytoplasm, and whole cells were detected through WB. Three biological replicates were performed for each sample. *, *p* < 0.05.

**Figure 6 ijms-25-00353-f006:**
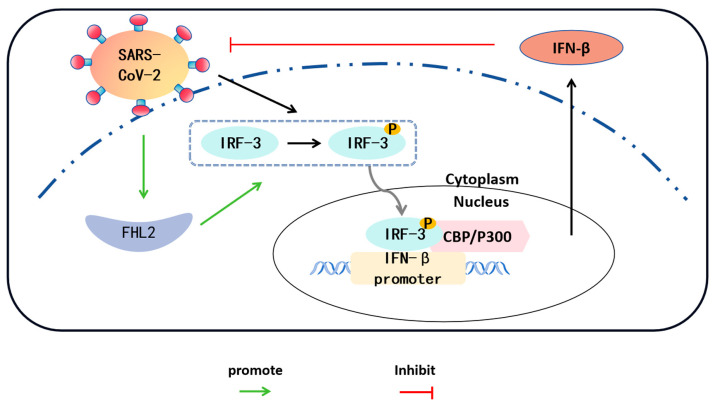
Interactions between SARS-CoV-2/*FHL2*/IRF-3. SARS-CoV-2 infection of Caco2 cells results in the phosphorylation of IRF-3 in the cytoplasm. The phosphorylated IRF-3 translocates to the nucleus and forms a complex with the coactivator CBP/P300, exhibiting transcriptional activity. Phosphorylated IRF-3 binds to the *IFN-β* promoter, leading to an increase in *IFN-β* expression. *IFN-β*, in turn, inhibits the replication of SARS-CoV-2. Concurrently, the expression level of *FHL2* in cells also elevates with SARS-CoV-2 infection. *FHL2* promotes the expression and phosphorylation of IRF-3, thereby enhancing the transcription of *IFN-β*.

**Table 1 ijms-25-00353-t001:** Primers in this study.

*Class*	Primer	Sequence (5′-3′)	Gene Name
*PCR*	*FHL2*F	GAATTCATGACTGAGCGCTTTGACTGCCA	*FHL2*
	*FHL*R	CTCGAGTCAATGGTGATGGTGATGGTGGATGTCTTTCCCACA	
	*M*F	GGTACCATGGCAGATTCCAACGGTACTATTACCG	*M*
	*M*R	CTCGAGTTACTTTTCGAACTGCGGGTGGCTCCACTGTACAAGCAAAGC	
	*N*F	GGTACCATGTCTGATAATGGACCCCAAAATCAGCG	*N*
	*N*R	TCTAGATTACTTTTCGAACTGCGGGTGGCTCCAGGCCTGAGTTGAGTC	
*qRT*- *PCR*	q*FHL2*F	CTGCCACCATTGCAACGAAT	*FHL2*
	q*FHL2*R	AGACAAGTCCTTGCAGTCACA	
	q*GAPDH*F	CTTTGGTATCGTGGAAGGACTC	*GAPDH*
	q*GAPDH*R	GTAGAGGCAGGGATGATGTTCT	
	q*IFN*-*β*F	TCTCCTGTTGTGCTTCTCCAC	*IFN*-*β*
	q*IFN*-*β*R	GCCTCCCATTCAATTGCCAC	
	q*IRF*-*3*F	AGAGGCTCGTGATGGTCAAG	*IRF*-*3*
	q*IRF*-*3*R	AGGTCCACAGTATTCTCCAGG	
	qNF	GGGGAACTTCTCCTGCTAGAA	N
	qNR	CAGACATTTTGCTCTCAAGCTG	

## Data Availability

All data generated or analyzed during this study are included in this published article and Appendix A.

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
