# Peer review of "FHL2 Inhibits SARS-CoV-2 Replication by Enhancing IFN-β Expression through Regulating IRF-3"

_ijms, 2023, doi:10.3390/ijms25010353_

Round 1

Reviewer 1 Report

Comments and Suggestions for Authors

Xu et al. have investigated the restriction of SARS-CoV-2 replication by FHL2 and report that the restriction factor acts independent of cell type or virus strain. This is a well performed study that additionally provides some insight into the mechanism of restriction. I have listed below some corrections that I think will improve the accuracy of the manuscript.

Corrections:

-       Authors should be more specific in the results section e.g. Lines 118 and 145 and 240, regarding what they are measuring. Authors should refer specifically to measurement of viral N protein levels by western blotting rather than just describing it as increased viral “replication”/ “proliferation of virus strains”.

-       Results Section 2.2 – Currently no reference to their findings with BA.1 strain.

-       Figure 2E/F: Can the authors compare the fold change in viral copies with/without shRNA-FHL2 between WT virus and BA.1? This may reveal if FHL2 is more highly restrictive to one strain compared to the other.

-       Lines 124/125 and other places: Authors state that FHL2 effect is “independent of virus strains and cell lines”. This statement is too broad given that only WT SARS-CoV-2 and BA.1 were tested and only in two cell lines were used. The BA.1 findings shown in Figure 2C/D/E are also not described in the text. Authors showed that FHL2 effects “replication” (viral N protein levels and viral RNA copies) of WT and BA.1 but the evidence they present does not rule our differential restriction between the viruses (the fold change analysis requested above may help). Other strains of SARS-CoV-2 were also not tested. Therefore, I think that the arguments that FHL2 action is independent of virus strain or cell type should be toned down. 

-       Figure 2C/D: Authors should include densitometry to see if the effects on viral N protein levels are dependent on the MOI. i.e. is there a does dependent effect?  

-       Lines 269, 270: Authors acknowledge a limitation of the study is that others have looked at restriction in different cell types and with different strains, this is further reason to remove arguments that FHL2 action is completely independent of virus strain/ cell type otherwise these differences would not matter.

-       Methods (Nuclear/Cytoplasmic Extraction): Please provide more details on the methodology used and controls demonstrating enrichment in the samples.

-       Have the authors tested if the antiviral activity of FHL2 extends to other coronaviruses?

-       Have the authors tested the activity of FHL2 e.g. by knockdown and infection in cells known to be infected by the virus in vivo such as lung cells?

-       From their analysis of the proteomic data, what other hits were found other than FHL2 and can these be shown (perhaps in a table?)

Comments on the Quality of English Language

The manuscript could do with careful editing/refining in places to ensure that the results are being described as specifically as possible. 

Author Response

Thank you very much for your time involved in reviewing the manuscript and your very encouraging comments on the merits.

We responded to your comments point by point. Please see the attachment for details.

Reviewer 2 Report

Comments and Suggestions for Authors

This study aimed to evaluate the role of FHL2 in SARS-CoV-2 infection in vitro.

Comments

1.      Consider drawing a figure where interactions between SARS-CoV-2/FHL2/IRF-3 would be shown. It will help to understand the role of FHL2.

2.      Line 13 - …site, mediates…

3.      Line 41 – After RNA release from endosomes, it is recognized by…

4.      Line 76 – …SARS-CoV-2 is associated with…

5.      Line 87 – Consider rephrasing the whole sentence.

6.      Line 89 – remains unclear whether FHL2 was the only one meeting the screening criteria.

7.      Line 90 – “infected cells were detected by WB” this sentence must be rephrased. I believe that collected cells were used to detect FHL2.

8.      Line 98 - …of N protein led to…

9.      Figure captions must be improved. Consider clarifying details, increasing font size (1A-1C).

10.  Line 115 – …that knockdown of FHL2 resulted in a significant decrease in FHL2 expression…

11.  Line 136 – in multiple places: viral N protein cannot be detected by qRT-PCR.

12.  Figure 2G – it would be interesting if you provided data on virus titer/qPCR for 0 hours p.i. to demonstrate whether the knockdown affected virus attachment along with its replication.

13.  Figure 3A-B – level of FHL2 expression does not seem to be higher in FHL2-his cells especially in 293T cells.

14.  Figure 3- consider providing data on TCID50 in FHL2-his/WT cells after infection.

15.  Gene names should be italicized. Check throughout the manuscript.

16.  Line 172 - Remains unclear why for evaluation of the effect of FHL2 knockdown you decided to use RNA extracted but not whole virus.

17.  Line 216 – rephrase the sentence.

18.  Line 218 – Does it mean that authors do not believe in effective vaccination against SARS-CoV-2. Needs to be clarified.

19.  Line 233. Consider double check if you study is actually the first one describing SARS-CoV-2 and FHL2.

Comments on the Quality of English Language

Scientific soundness must be improved

Author Response

Thank you very much for your time involved in reviewing the manuscript and your very encouraging comments on the merits.

We responded to your comments point-by-point. Please see the attachment for details.

Round 2

Reviewer 1 Report

Comments and Suggestions for Authors

I thank the authors for responding to my comments and making the relevant adjustments. The manuscript is now suitable for publication.

Comments on the Quality of English Language

Minor corrections required in copy editing 

Author Response

Dear Reviewer,

Thank you very much for your time involved in reviewing the manuscript and your very encouraging comments on the merits.

General Comments:

I thank the authors for responding to my comments and making the relevant adjustments. The manuscript is now suitable for publication.

Response :

The entire manuscript has been polished further by the native language company.

We would like to take this opportunity to thank you for all your time involved and this great opportunity for us to improve the manuscript. We hope you will find this revised version satisfactory.

Sincerely,

The Authors
